

# Systematic and meta-based evaluation of the relationship between the built environment and physical activity behaviors among older adults

Yanwei You[1,2,*], Yuquan Chen[3,*], Qi Zhang[4], Xiaojie Hu[5], Xingzhong Li[6,7], Ping Yang[8], Qun Zuo[9] and Qiang Cao[10,11]

[1] Division of Sports Science and Physical Education, Tsinghua University, Bejing, China
[2] School of Social Sciences, Tsinghua University, Beijing, China
[3] Institute of Medical Information/Medical Library, Chinese Academy of Medical Sciences & Peking Union Medical College, Beijing, China
[4] Taishan University, Taian, China
[5] Shanghai University of Traditional Chinese Medicine, Shanghai, China
[6] Zhedong Orthopedic Hospital, Ningbo, China
[7] Current Affiliation: Orthopedics Department, PLA Rocket Force Characteristic Medical Center, Beijing, China
[8] Department of Epidemiology, Johns Hopkins Bloomberg School of Public Health, Baltimore, United States
[9] College of Public Health, Hebei University/Hebei Key Laboratory of Public Health Safety, Baoding, China
[10] Department of Earth Sciences, Kunming University of Science and Technology, Kunming, China
[11] School of Pharmacy, Macau University of Science and Technology, Macau, China
* These authors contributed equally to this work.

Corresponding authors
Qun Zuo, zuoqun2006@126.com
Qiang Cao, 2918292861@qq.com

## ABSTRACT

**Objectives:** Existing assertions about the relationship between various factors of the built environment and physical activity behaviors are inconsistent and warrant further exploration and analysis.

**Methods:** This study systematically searched PubMed, Embase, Web of Science, Scopus, the Cochrane Library and Google Scholar for the effect of the built environment on the physical activity behaviors of older adults. R software was used to calculate the meta-estimated odds ratio with a 95% confidence interval. Simultaneously, the quality of included studies was evaluated using an observational study quality evaluation standard recommended by American health care quality and research institutions.

**Results:** A total of 16 original researches were included in this meta-analysis and eight factors of the built environment were evaluated. These factors which ranked from high to low according to their impact were traffic safety (OR = 1.58, 95% CI [1.14–2.20]), destination accessibility (OR = 1.24, 95% CI [1.06–1.44]), aesthetics of sports venues (OR = 1.21, 95% CI [1.07–1.37]), virescence of sports venues (OR = 1.14, 95% CI [1.06–1.23]), building density (OR = 1.07, 95% CI [1.02–1.13]). Additionally, it seemed that there was no potential association between mixed land use (OR = 1.01, 95% CI [0.92–1.10]), the quality of pedestrian facilities (OR = 1.00, 95% CI [0.92–1.08]) or commercial facilities (OR = 0.94, 95% CI [0.88–1.00]) and physical activity behaviors of older adults.

**Conclusions:** The built environment has been found to exhibit a significant relationship with the physical activity behaviors of older adults. It is proposed that factors such as traffic safety, destination accessibility, aesthetics of sports venues, virescence of sports venues, and building density be given more consideration when aiming to promote physical activity levels among older adults.

## INTRODUCTION

The swift evolution of social economy and science and technology, such as human transportation and travel, has indeed made production and living more convenient. However, it has concurrently contributed to widespread physical inactivity, an influential factor leading more and more individuals towards sub-optimal health or even a disease state (*Richards, McDonough & Fu, 2017*). The survey results of the World Health Organization (WHO) indicate that physical inactivity has become the fourth leading risk factor for death in the world. The annual death of nearly three million people can be attributed to physical inactivity, and this number has grown rapidly in the past 10 years (*Cerin et al., 2017*). For the older population in particular, not only do physiological declines and the incidence of chronic diseases increase, but also psychological problems such as cognitive impairment, depression and sleep disturbances become more evident (*You et al., 2022b*, *2022a*, *2023a*). Regular physical activity (PA) and a healthy lifestyle can improve both physical and mental function, and further ameliorate quality of life and social adaptability (*You et al., 2023b*, *2023f*, *2023c*). PA is deemed to be especially beneficial for older adults, making it a crucial strategy for maintaining a high cost-benefit ratio of health for this demographic.

Consequently, understanding the factors influencing the physical activity behavior (PAB) of older adults becomes imperative to effectuate successful interventions and facilitate healthy aging. In 1968, *Barker (1968)* created ecological psychology. After long-term observation of children in their daily environment, they proposed that the environment had a direct effect on their behaviors (*Peralta et al., 2022*; *Wold & Samdal, 2012*). In 1977, *Bronfenbrenner (1977)* put forward the theory of ecological system for the first time in combination with the thought of ecological psychology, dividing the individual external factors into two levels according to the close and far individuals, namely, the micro and macro system, and dividing the impact and intervention level of the ecological model into the individual internal (individual itself) level and the individual external (external environment) level (*Sallis & Owen, 2002*). In 1979, inspired by Kurt Lewin's topological field theory (*Adelman, 1993*), *Bronfenbrenner (1979)* described the ecological environment as a nested structure of Russian doll like systems, and thus divided four environmental components: microsystem, mesosystem, exo-system and macrosystem, and proposed an ecological model of individual development (*Dijkstra et al., 2022*). In 1988, *McLeroy et al. (1988)* classified the factors that determine individual behaviors into five categories, namely,

personal factors, interpersonal relationships, institutional factors, community factors and public policies, from proximal to distal, and proposed a famous ecological model of health promotion. In 1992, based on *Bronfenbrenner*'s *(1977)* theory, *Wachs (1992)* introduced social support, physical characteristics and high-level regulatory variables to build a Structural Model of the Environment for the development of children. *Wachs (1992)* put forwards the interaction between various factors and indirectly affected individual behavior through high-level regulatory variables, which became the theoretical basis of *Spence & Lee*'s *(2003)* ecological model of PABs (*Peralta et al., 2022*). *Spence & Lee (2003)* took individual biological factors into consideration on the basis of *Wachs (1992)* and built a more comprehensive ecological model of PABs, which is used to explain those behaviors in the population of youngsters (*Cerin et al., 2017*; *Ewing & Cervero, 2001*). The schematic diagram of the ecological model of health promotion is shown in Appendix A.

One of the guiding principles of the ecological model of health promotion for the study of PA of older adults indicates that the built environment plays an important role in inhibiting or promoting the occurrence and development of PAB, but there is still a lack of systematic evaluation in this regard (*Franco et al., 2015*; *McPhee et al., 2016*; *Molanorouzi, Khoo & Morris, 2015*; *Stults-Kolehmainen & Sinha, 2014*). In a broad sense, the built environment includes all kinds of buildings and places constructed with humans as the center (parks, schools, gyms, commercial areas, *etc.*), while artificial adjustment or policy change can affect such as comprehensive land management and utilization, population density, *etc.*, which also belong to the concept of that (*Hawkesworth et al., 2018*; *Zhong et al., 2022*). In addition, based on a narrow perspective, the built environment mainly includes five dimensions of indicators: density, mixing degree, block design, public transport proximity and destination accessibility, which are collectively referred to as 5Ds indicators (*Miralles-Guasch et al., 2019*). Furthermore, with regard to the insight into city planning, the built environment can be divided into three aspects: land use, transportation system and urban design (*Zhong et al., 2022*; *Alidadi & Sharifi, 2022*).

Early assessment methods of the built environment were mainly qualitative, such as photo evaluation, interview, questionnaire, *etc.*, *Li et al. (2022)*. In recent years, with the maturity of GIS, RS, GPS and other technologies, quantification has been achieved (*Pontin et al., 2022*). There were three main methods of evaluation for built environment: (i) Subjective evaluation method; (ii) objective scanning method; and (iii) using GIS technology to analyze existing geographical data (*Li et al., 2022*; *Pontin et al., 2022*; *Gába et al., 2022*). Evaluation scales for the built environment are shown in Appendix B.

However, the relationship between the built environment and the PABs of older adults has received widespread attention from the academic community, but there have not been more consistent research results though (*Cerin et al., 2017*; *Hawkesworth et al., 2018*; *Zhong et al., 2022*). *Hanibuchi et al. (2011)* conducted a study that revealed a positive correlation between population density and parks or greenspaces with the frequency of physical activity among older individuals in Japan. Similarly, *Smith et al. (2017)* compared park and greenspace use between older men and women, finding that women used these spaces less frequently, and parks and greenspaces primarily promoted recreational physical activity in older men. Regarding traffic, *Sallis et al. (2013)* proposed that enhancing traffic

safety could encourage outdoor walking among older adults, while *McGinn et al. (2007)* argued that traffic safety was not a significant factor affecting both traffic-related and recreational physical activity. Moreover, to the best of our knowledge, the relationship between the built environment and PABs has been well explored in children and teenagers (*McGrath, Hopkins & Hinckson, 2015*; *Quigg et al., 2010*; *Rodríguez et al., 2012*), while the evidence between such exposures in older adults was limited. In view of the above, this article conducted a systematic evaluation in which a meta-analysis was used to quantitatively synthesize multiple original research results.

As for the significance of this study, our research will provide insights into the nuances of the relationship between the built environment and PAB in older adults. This has implications not only for the health of older adults but also for policy makers, city planners, and community leaders seeking to promote active lifestyles among the aging population. Secondly, by examining the inconsistencies in previous research, our work can serve as a foundation for future research in this area. Ultimately, our findings may help foster strategies and interventions that promote healthier behaviors and environments, contributing to the broader objective of enhancing public health and quality of life.

## METHODOLOGY

Meta-analysis was applied in this study. Compared with traditional literature review or the emerging bibliometric analysis, systematic review and meta-analysis had a relatively broad horizon of the current hotspots and could quantitatively reflect the research status in the field (*Chen et al., 2022*; *You et al., 2021b*, *2021a*). This systematic review was conducted in accordance with the Preferred Reporting Items for Systematic Reviews and Meta-Analysis Protocols guidelines (*BMJ, 2016*) and was registered with the International Prospective Register of Systematic Reviews (PROSPERO, registration number: CRD42022342176).

### Search strategy

Cross-sectional studies on the effect of the built environment on PAB of older adults published in electronic database were searched by computer, including PubMed, Embase, Web of Science, Scopus, the Cochrane Library and Google Scholar. At the same time, experts in the relevant field were consulted in order to obtain additional information and obtain potential literature. The retrieval time limit was set from the inception of the database to January 01, 2022. The search strategy was based on a combination of: 'physical activity', 'physical exercise', 'sports', 'elderly', 'older adults', 'environment', 'built environment', 'aesthetics', 'pedestrian facilities', 'commercial facilities', 'density', 'accessibility', 'traffic safety', 'mixed land use', 'urban design', 'neighborhood characteristics', 'public spaces', and 'urban planning', *etc*. We used Boolean operators (within each axis, we combined keywords with the "OR" operator to expand the search, and we then linked the search strategies for the two axes with the "AND" operator to narrow the search). A sample of the search strategy in PubMed, developed using a combination of MeSH terms and free texts is provided in Appendix C.

## Inclusion and exclusion criteria

According to the relevant references, this study defines the PABs of older adults as activities with certain intensity, frequency and duration for older adults aged 60 and above to improve their physical and mental health, quality of life, social adaptation, *etc.*, in their spare time.

Inclusion criteria: (1) Types of studies: original cross-sectional studies. (2) Types of participants: older adults without cognitive impairment. (3) The outcome of influencing factors measures: the OR value of different levels of built environment (including one or more of the aesthetic degree, greening degree, quality of pedestrian facilities, commercial facilities, housing density, destination accessibility, traffic safety, and diversity of mixed land use, *etc.*) and standardized partial regression coefficient (or 95% *CI* of *OR* value) were reported or could be further calculated based on univariate or multivariate analysis.

Literature exclusion criteria: (1) Case report, review, systematic evaluation, and meta-analysis. (2) Repeated published and poor-quality literature. (3) The information is incomplete, and the relevant data cannot be obtained or is missing.

## Literature screening process

Firstly, the title information of relevant literature was retrieved through the retrieval strategy, and Endnote X9 software was used for literature management. After duplication of the included publications, two reviewers (X.H. and X.L.) read the title and abstract for preliminary screening according to the inclusion and exclusion criteria and then read the full text to judge the qualification. A third reviewer (Y.Y. and Y.C.) resolved disagreements about the inclusion criteria. For the qualified literature finally selected, two parallel groups (X.H., X.L. and P.Y.) independently extracted the research data and made records, including the first author, survey time, survey area, sampling method, sample size, proportion of male, and indicators reported of built environment, *etc.*

## Quality assessment

Two independent reviewers, X.H. and X.L., assessed the risk of bias in the study and cross-checked their evaluations. In cases where the two reviewers had differing opinions, the final conclusion was reached through discussion involving the third reviewer, Y.Y., and Y.C. The quality of cross-sectional studies was evaluated using 11 items from the observational study quality evaluation standard recommended by American healthcare quality and research institutions (*Miller, Vandome & Mcbrewster, 2002*). Each item was assigned a score, with a total possible score of 11 points. Studies were classified as low-quality if they scored 0 to 3 points, medium quality if they scored 4 to 7 points, and high quality if they scored 8 to 11 points. The risk of bias (ROB) in the original study was determined based on the quality assessment results.

## Data analysis

Data analysis was conducted using the meta package in R software (version 4.0.3; *R Core Team, 2020*). To assess the effect of the built environment, we extracted risk summary measures (odds ratio) with 95% confidence intervals (CI) for the influencing factors of

interest as provided in the included studies. The generic inverse variance method was employed for this analysis. To ensure the accuracy of the multivariate analysis and maintain the principle of comprehensive retrieval, we prioritized the research results based on the strategy of multivariate analysis, and then incorporated the results of the univariate analysis. In cases where the combined effect quantity OR = 1 or the 95% CI included the invalid vertical line (abscissa scale value of 1), it indicated no relationship between the built environment factor and the outcome variable.

When the combined effect (OR) was greater than 1 and the lower limit of the 95% CI was also greater than 1, the interval appeared to the right of the invalid line in the forest plot. This indicated a significant positive correlation with the outcome indicators, which we have referred to as 'positively associated factors'. Conversely, when the combined effect (OR) and the upper limit of the 95% CI were both less than 1, and the interval appeared to the left of the invalid line, it indicated a significant inverse correlation with the outcome indicators, which we refer to as 'negatively associated factors'. The Cochrane $Q$ test and $I^2$ value were used to test whether there was heterogeneity among studies (*Tu & Greenwood, 2012*). According to the Meta-analysis of Observational Studies in Epidemiology guideline (*Higgins et al., 2003*), if $P > 0.10$ and $I^2 \leq 40\%$, it indicated no statistical heterogeneity among the research results, and the fixed effect model was applied to analyze the results. If $P \leq 0.1$ and $I^2 > 40\%$, the random effect model was used for meta-analysis. Simultaneously, publication bias was evaluated using Egger's test and significant clinical heterogeneity was treated by subgroup analysis. Furthermore, sensitivity analysis was performed by excluding low-quality studies or choosing fixed effect model. When the point estimate deviated from the main analysis result by more than or equal to 20%, it was considered that the result was greatly affected by low-quality research or effect model selection. The difference was considered as significant when the $P$ value for the comparison was less than 0.05.

## Patient and public involvement

Patients and/or the public were not involved in the design, or conduct, or reporting, or dissemination plans of this research.

# RESULTS

## Literature search results

A total of 1,088 articles were obtained from various databases and references recommended by experts. A total of 408 duplicate articles were eliminated by using Endnote *X9* software, and 196 irrelevant articles were eliminated by reading titles and abstracts. Subsequently, type of review studies, documents with inconsistent research objects and incomplete data information were excluded by reading the full text. Finally, a total of 16 articles were included for the qualitative and quantitative analysis (*Hawkesworth et al., 2018*; *Miralles-Guasch et al., 2019*; *Hanibuchi et al., 2011*; *Koohsari et al., 2020*; *Barnett et al., 2016*; *Gao et al., 2015*; *de Sa & Ardern, 2014*; *Zhang et al., 2014*; *Cerin et al., 2013*; *Ribeiro et al., 2013*; *Tsunoda et al., 2012*; *Witten et al., 2012*; *Carlson et al.,*

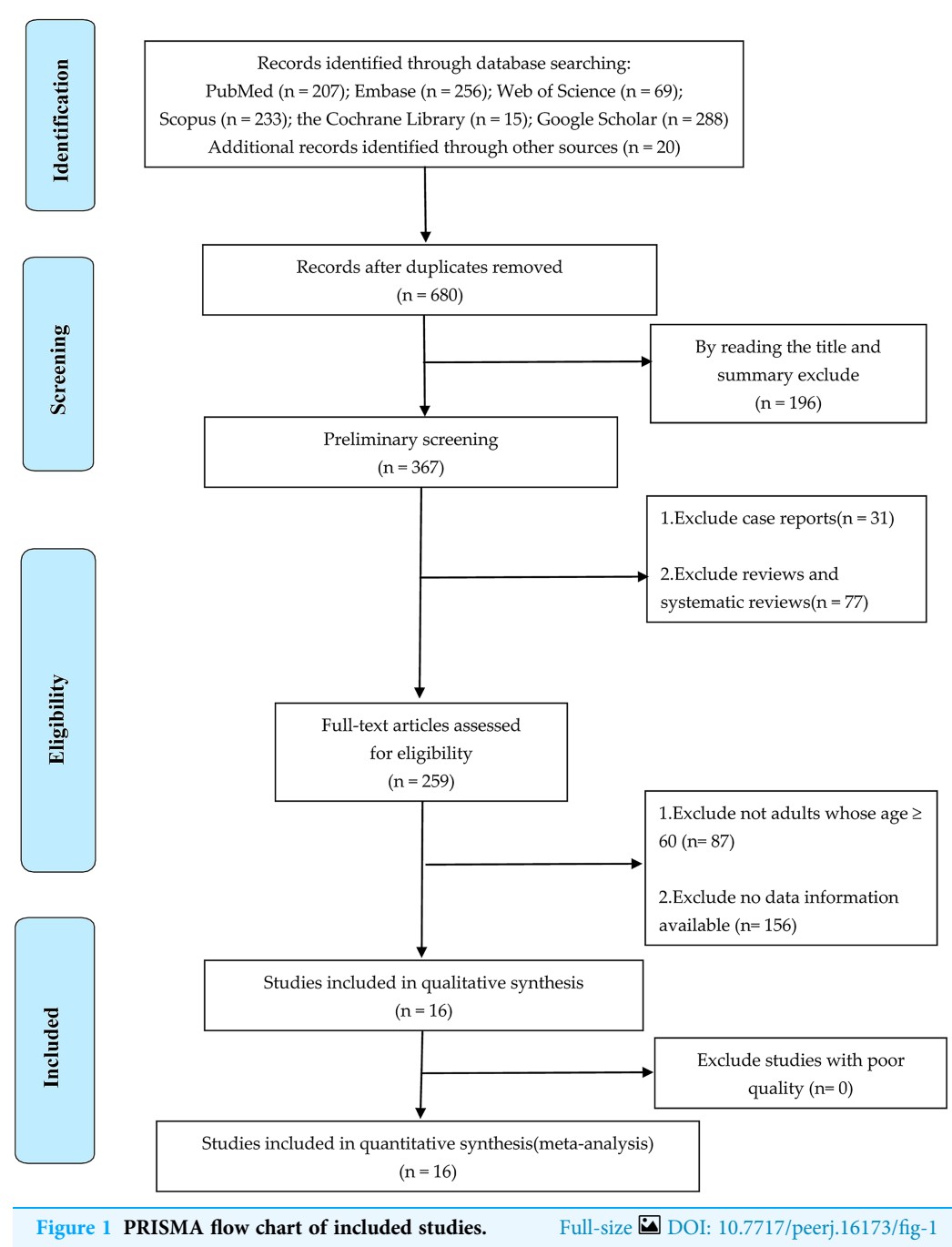

**Figure 1 PRISMA flow chart of included studies.**

*2012*; *Corseuil et al., 2011*; *Parra et al., 2010*; *Satariano et al., 2010*). See Fig. 1 for a detailed process.

## Data extraction results

Characteristics of 16 included original studies on the effect of the built environment on PABs of older adults are shown in Table 1. These studies were performed in 11 different nations or regions, including 29,113 participants. Among them, seven studies were conducted in Asia (*Hanibuchi et al., 2011*; *Koohsari et al., 2020*; *Barnett et al., 2016*; *Gao*

**Table 1 Characteristics of 16 included studies on the effect of the built environment on physical activity behaviors of the elderly.**

| Study ID | First author | Publication year | Survey area | Sample size | Age (years) | Proportion* | Outcomes |
|---|---|---|---|---|---|---|---|
| 1 | Koohsari (*Gao et al., 2015*) | 2020 | Japan | 314 | 65–84 | 61.8 | ⑥⑦ |
| 2 | Carme (*Li et al., 2022*) | 2019 | Spain | 269 | >65 | 55.6 | ②⑤ |
| 3 | Sophie (*Miralles-Guasch et al., 2019*) | 2018 | England | 1,433 | 69–92 | 55.5 | ①②④⑥ |
| 4 | Anthony (*de Sa & Ardern, 2014*) | 2016 | Hong Kong | 909 | 76 ± 6 | 34 | ①⑤⑥⑧ |
| 5 | Junling Gao (*Zhang et al., 2014*) | 2015 | China | 2,783 | >65 | 41.1 | ① |
| 6 | Eric de Sa (*Cerin et al., 2013*) | 2014 | Canada | 1,158 | >60 | 50.4 | ②⑥ |
| 7 | Yi Zhang (*Ribeiro et al., 2013*) | 2014 | China | 4,308 | >60 | 47.5 | ②④⑤ |
| 8 | Ester Cerin (*Tsunoda et al., 2012*) | 2013 | Hong Kong | 484 | >65 | 42 | ①② |
| 9 | Ana Isabel (*Witten et al., 2012*) | 2013 | Portugal | 580 | >65 | 42.1 | ⑥⑦ |
| 10 | K. Tsunoda (*Carlson et al., 2012*) | 2012 | Japan | 421 | 73.3 ± 5.3 | 47.5 | ①②③④⑥ |
| 11 | Witten (*Corseuil et al., 2011*) | 2012 | New Zealand | 1,806 | 55–65 | 57.2 | ①⑥⑦⑧ |
| 12 | Carlson (*Parra et al., 2010*) | 2012 | U.S.A | 718 | 74.4 ± 6.3 | 46.9 | ①④⑤ |
| 13 | Corseuil (*Satariano et al., 2010*) | 2011 | Brazil | 1,666 | >60 | 36.1 | ②④ |
| 14 | Tomoya (*Sallis et al., 2013*) | 2011 | Japan | 9,414 | >65 | 48 | ①⑤⑥⑦ |
| 15 | Parra (*Barcelona et al., 2022*) | 2010 | Columbia | 1,966 | 70 ± 7.7 | 37.5 | ②⑧ |
| 16 | Satariano (*Chen & Zuo, 2019*) | 2010 | U.S.A | 884 | >65 | 23.4 | ③⑥ |

Notes:
* Proportion: namely the male one.
Outcomes: ①, aesthetics (sports venues); ②, virescence (sports venues); ③, traffic safety; ④, quality of pedestrian facilities; ⑤, commercial facilities; ⑥, building density; ⑦, destination accessibility; ⑧, diversity of mixed land use.

*et al., 2015*; *Zhang et al., 2014*; *Cerin et al., 2013*; *Tsunoda et al., 2012*), three studies in Europe (*Hawkesworth et al., 2018*; *Miralles-Guasch et al., 2019*; *Ribeiro et al., 2013*), three studies in North America (*de Sa & Ardern, 2014*; *Carlson et al., 2012*; *Satariano et al., 2010*), two studies in South America (*Corseuil et al., 2011*; *Parra et al., 2010*) and only one in Oceania (*Witten et al., 2012*). Moreover, of the studies with over 50% of male participants, there were only five (*Hawkesworth et al., 2018*; *Miralles-Guasch et al., 2019*; *Koohsari et al., 2020*; *de Sa & Ardern, 2014*; *Witten et al., 2012*).

Table 2 demonstrates the quality evaluation of original literature, including 10 high-quality studies, six medium-quality studies and no low-quality studies. The average score of the quality of overall studies was 8.06, and the standard deviation was 1.48. Summary and traffic light plot of the risk bias assessment of all studies are shown in Fig. 2. After quality evaluation, it could be seen that the overall quality of the original study was satisfactory. The literature included in the final study could be directly analyzed qualitatively and quantitatively.

## Meta-estimated results of effects of built environment
### Traffic system
The results of meta-estimated *OR* value was 1.24 (95% CI [1.06–1.44]) between the destination accessibility and PAB, which indicated that it is a significant positively associated factor and is presented in Fig. 3A. In addition, the relationship between traffic safety and PAB in older adults showed a similar trend, with a meta-estimated *OR* value of 1.58 (95% CI [1.14–2.20]), which could be attributed to a positively associated factor.

**Table 2 Quality evaluation results of 16 included studies of the effect of the built environment on physical activity behaviors of the elderly.**

| Study ID | PY* | First author | D1 | D2 | D3 | D4 | D5 | D6 | D7 | D8 | D9 | D10 | D11 | Overall |
|---|---|---|---|---|---|---|---|---|---|---|---|---|---|---|
| 1 | 2020 | Koohsari | 1 | 1 | 1 | 1 | 1 | 0 | 1 | 1 | Unclear | 1 | 1 | 9 |
| 2 | 2019 | Carme | 1 | 0 | 1 | 1 | Unclear | 0 | 0 | 1 | Unclear | 1 | 1 | 6 |
| 3 | 2018 | Sophie | 1 | 0 | 1 | 1 | 0 | 0 | 1 | 1 | Unclear | 1 | 1 | 7 |
| 4 | 2016 | Anthony | 1 | 1 | 1 | 1 | Unclear | 1 | 1 | 1 | Unclear | 1 | 1 | 9 |
| 5 | 2015 | Junling Gao | 1 | 1 | 0 | Unclear | 0 | 1 | 1 | 0 | Unclear | 1 | 1 | 6 |
| 6 | 2014 | Eric de Sa | 1 | 1 | 1 | 1 | 1 | 1 | 1 | Unclear | 1 | 1 | 1 | 10 |
| 7 | 2014 | Yi Zhang | 1 | 0 | 0 | 1 | 1 | 0 | 0 | 1 | Unclear | 1 | 1 | 6 |
| 8 | 2013 | Ester Cerin | 1 | 1 | 1 | 1 | Unclear | 1 | 1 | Unclear | 1 | 1 | 1 | 9 |
| 9 | 2013 | Ana Isabel | 1 | 1 | 1 | 1 | Unclear | 1 | 1 | Unclear | 1 | 1 | 1 | 9 |
| 10 | 2012 | K. Tsunoda | 1 | 1 | 1 | 1 | Unclear | 1 | 1 | 1 | 1 | 1 | 1 | 10 |
| 11 | 2012 | Witten | 1 | 0 | 1 | 1 | Unclear | 0 | 0 | 1 | 0 | 1 | 1 | 6 |
| 12 | 2012 | Carlson | 1 | 0 | 0 | 1 | 1 | 0 | 1 | 1 | 1 | 1 | 1 | 8 |
| 13 | 2011 | Corseuil | 1 | 1 | 1 | 1 | 1 | 1 | 1 | Unclear | 1 | 1 | 1 | 10 |
| 14 | 2011 | Tomoya | Unclear | 0 | 1 | Unclear | 1 | 1 | 1 | 1 | Unclear | 1 | 1 | 7 |
| 15 | 2010 | Parra | 1 | 0 | 1 | 1 | 0 | 1 | 1 | 1 | Unclear | 1 | 1 | 8 |
| 16 | 2010 | Satariano | 1 | 1 | 1 | 1 | Unclear | 1 | 0 | 1 | 1 | 1 | 1 | 9 |

**Notes:**
D1, Define the source of information (survey, record review); D2, list inclusion and exclusion criteria for exposed and unexposed subjects (cases and controls) or refer to previous publications; D3, indicate time period used for identifying patients; D4, indicate whether or not subjects were consecutive if not population-based; D5, indicate if evaluators of subjective components of study were masked to other aspects of the status of the participants; D6, describe any assessments undertaken for quality assurance purposes (*e.g.*, test/retest of primary outcome measurements); D7, explain any patient exclusion from analysis; D8, describe how confounding was assessed and/or controlled; D9, if applicable, explain how missing data were handled in the analysis; D10, summarize patient response rates and completeness of data collection; D11, clarify what follow-up, if any, was expected and the percentage of patients for which incomplete data or follow-up was obtained.
* PY, publication year.

The forest plot is presented in Fig. 3B. It should be noted that the results of both of these two outcome indicators were explained by the random effect model because the $I^2$ was greater than 40% and $P$ value was less than 0.1.

### *Urban design*

For the effects of aesthetics of sports venues on the PAB of older adults, its *OR* value is 1.21 (95% CI [1.07–1.37]) and presented in Fig. 3C. Despite the relatively large heterogeneity in the study and after performing a sensitivity analysis by excluding the included studies one by one, the study conducted by *Cerin et al. (2013)* might be one of the sources of heterogeneity and, if excluded, the final fixed effect model result would be 1.24 (95% CI [1.15–1.34]). In addition, with regard to virescence, the *OR* value was 1.14 (95% CI [1.06–1.23]) with PAB of older adults and its forest plot is presented in Fig. 3D. Simultaneously, the *OR* value between the effect of pedestrian facilities on PAB of older people was 1.00 (95% CI [0.92–1.08]), which indicated this indicator may not be associated, as shown in Fig. 3E. Last but not least, the meta-estimated results showed that the final *OR* value was 0.94 (95% CI [0.88–1.00]) for the effects of aesthetics of commercial facilities on the PAB of older adults, and its forest plot is presented in Fig. 3F.

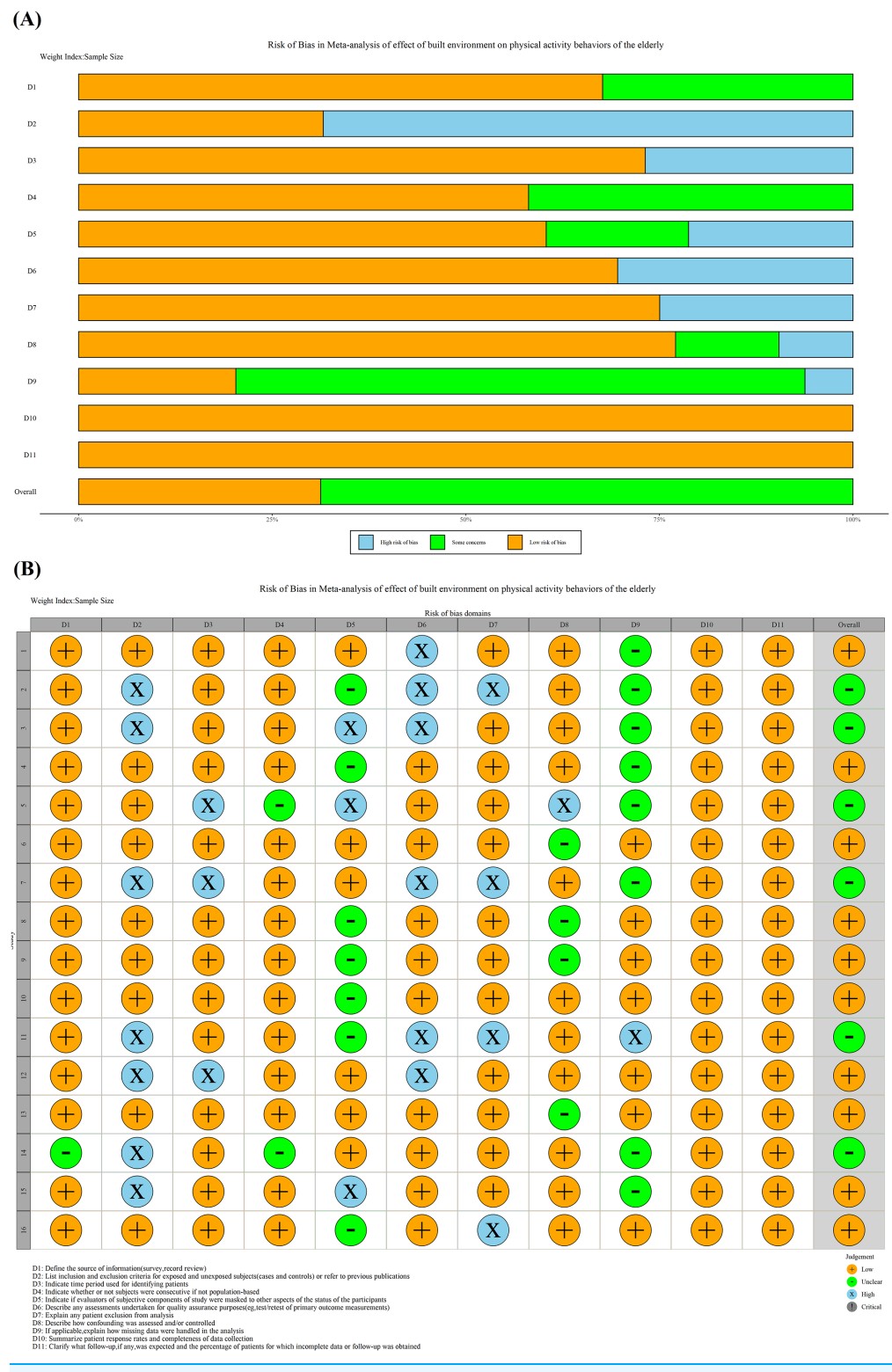

**Figure 2** **(A)** **Summary plot of risk bias assessment. (B) Traffic light plot of the risk bias assessment.**

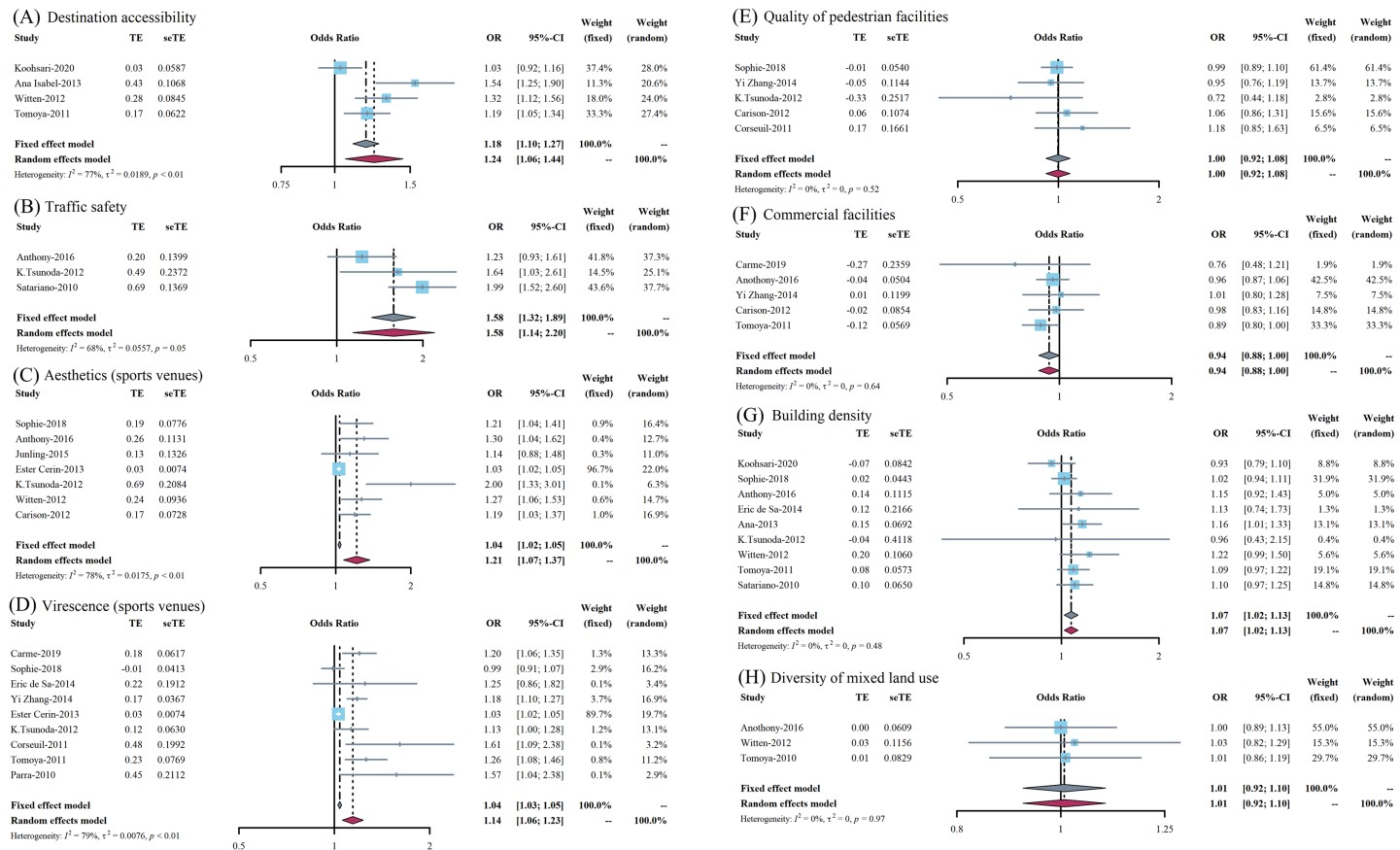

**Figure 3 (A–H) Forest plots of the effect on different outcome indicators of the built environment on physical activity behaviors of the elderly.**

### Land use

As for building density, the meta-estimated *OR* value is 1.07 (95% CI [1.02–1.13]) and presented in Fig. 3G, which indicated that it was a positively associated factor. However, the meta-estimated *OR* value between the diversity of mixed land use on PAB of older adults was 1.01 (95% CI [0.92–1.10]), suggesting that this indicator is not related to PAB, as shown in Fig. 3H.

### Summary results

Overall, many indicators of the built environment could promote the development of PAB. Based on the above results, these factors which ranked from high to low according to their impact were traffic safety (1.58), destination accessibility (1.36), aesthetics of sports venues (1.24), virescence of sports venues (1.14), building density (1.07), and it seemed that there was no potential association between mixed land use, the quality of pedestrian facilities or commercial facilities and PAB of older adults.

### Publication bias and sensitivity analysis

In our study, *Egger's* linear regression method was used to quantitatively evaluate publication bias. According to the test of publication bias in Table 3, only the meta-analysis

**Table 3 Evaluation results of publication bias using the method of Egger's test.**

| Outcome indicators | No.[*] | Standard error | t | P |
|---|---|---|---|---|
| Aesthetics (sports venues) | 7 | 0.34018204 | 6.5082 | 0.001279 |
| Virescence (sports venues) | 9 | 0.55555072 | 3.6192 | 0.08519 |
| Traffic safety | 3 | 0.1933918 | 0.030012 | 0.9809 |
| Quality of pedestrian facilities | 5 | 0.9719480 | −0.25491 | 0.8153 |
| Commercial facilities | 5 | 0.95331472 | −0.28683 | 0.7929 |
| Building density | 9 | 0.58919441 | 1.1724 | 0.2855 |
| Destination accessibility | 4 | 0.6411639 | 8.7115 | 0.07276 |
| Diversity of mixed land use | 3 | 0.04366358 | 12.137 | 0.05233 |

**Note:**
[*] The number of included studies.

of aesthetics of sports venues had a significant publication bias, while others did not. The fluctuations observed in different elements were less than 20% when transitioning between the fixed effect model and the random effect model. This indicates a relative stability in the estimation results.

# DISSCUSSIONS

A plethora of existing research has shown that, apart from the built environment, all other dimensions elucidated by the ecological model of health promotion have been confirmed to have either direct or indirect associations with PABs in older adults (*Barcelona et al., 2022*; *Chen & Zuo, 2019*). However, for many factors of the built environment, there has not been a systematic evaluation, and relatively consistent research results have not been conducted in academia (*Barcelona et al., 2022*; *Wylie et al., 2022*). To the best of our knowledge, this is the first study to evaluate the associations between diverse factors of the built environment and the PAB of older adults based on systematic review and meta-analysis. According to the analysis results, among the selected outcome indicators, traffic safety, destination accessibility, aesthetics, virescence and building density could promote the development of this kind of behavior. Traffic safety mainly focuses on objective traffic conditions, which refers to the condition or degree that activities can be carried out safely on the road and avoid personal injury or financial loss. *Tsunoda et al. (2012)*, has shown that increased traffic safety will effectively promote leisure time physical activity among older adults. However, this is contrary to *Inoue et al.*'s *(2011)* conclusion that there was no potential relationship between traffic safety and adult PA. These might be explained by the fact that different traffic conditions differed in diverse regions, or the age groups and cognitive conditions of the selected research subjects were various (*Amuzie et al., 2022*; *Yue et al., 2022*). Consequently, the government can improve the safety of regional traffic in order to promote older adults' PABs.

As for the aspect of urban design, firstly, *Su et al. (2014)*, indicated that the perception of aesthetics of sports venues has a strong correlation with the PABs of older adults, which was consistent with other scholars' research results (*Hawkesworth et al., 2018*; *Hanibuchi et al., 2011*; *Barnett et al., 2016*; *Gao et al., 2015*; *Carlson et al., 2012*). However, *Jia & Fu*

*(2014)* pointed out that there was no significant association between this factor and PABs of older adults. Despite the fact that we omitted the original study that resulted in the large heterogeneity in our study, the results still showed that PAB in older adults individuals could be contributed to by good aesthetic perception of sports venues (*Leonkiewicz & Wawrzyniak, 2022*). This might be related to differences in regional culture and customs and the study areas might also be the source of heterogeneity (*Robinson et al., 2022*). The reasons for this phenomenon could be further explored in future studies. Hence, based on the above mentioned, the government should focus on the aesthetics of buildings, better design the planning city and improve the aesthetic degree, for the purpose of promoting the PAB of older adults at a certain level (*Armada Martínez et al., 2021*; *Skervin et al., 2021*). Similarly, the virescence of sports venues could also promote older adults' PABs. In the surrounding environments where older adults live, if greening levels and densities were high or the green area was large, the frequency and duration of physical activity of older adults would increase (*Li, Hachenberger & Lemola, 2022*). This might be due to the extension of viewing green time for older adults because of greening, and thus the corresponding exercise time would also extend (such as walking) (*Nielsen et al., 2022*). Therefore, the management and planning department can promote the occurrence of PAB in older adults by improving the degree of regional greening, such as increasing the area of parkland space, so as to strengthen the physical fitness of older adults.

A study from the United States found that residents living in rural or low density housing areas (OR = 1.29, 95% CI [1.04–1.60]) were more likely to participate in physical activity than residents living in urban or high density housing areas (*OR* = 1.23, 95% CI [1.07–1.42]) (*Blackwell, Lucas & Clarke, 2014*), contrary to that of *Hawkesworth et al. (2018)*, *Barnett et al. (2016)*, and *de Sa & Ardern (2014)*. As shown in our summary results, there was indeed a correlation between the building density and the PAB of older adults, which suggested the government should take measures to increase the density of housing to encourage this behavior (*Yue et al., 2022*). As for the significant contradiction, there was a possibility that the relationship between building density and PABs of older adults existed a striking "threshold effect", namely when it did not exceed a certain threshold, it was consistent with the results of most studies, but when it was greater than this threshold, that was, excessive housing density would limit the older adults' PBA (*Abdelrahman et al., 2022*). However, more research was needed to determine whether this hypothesis was tenable. Furthermore, the essence of mixed land use diversity was to promote the sustainable development of the city by reducing land waste and energy consumption. However, our results indicated that there was no clear relationship between this indicator and PAB of older adults. When formulating corresponding policies, the relevant departments need not give priority to the impact of mixed land use. However, considering that there were relatively few research objects included, it was still necessary to further explore the relationship between two of them (*Sugiyama et al., 2021*).

Our results have also shown that there was no potential association between the quality of pedestrian facilities or commercial facilities and PAB of older adults (*Kim, Park & Kang, 2022*). Consequently, it could be concluded that there was an interaction between environmental factors and individual behaviors. Not all elements of the built environment

could have an independent impact on older adults's PAB though (*Amaya et al., 2022*). This might be explained by the fact that the complexity of the influencing factors and the uncertainty of their interactions are hidden. For example, the impact and role of commercial facilities might be hidden by the fact that when commercial facilities increased, traffic safety would also decline (*Xiao et al., 2022*). Thus, academia should carry out in-depth research on the impact mechanisms of PA for older adultse, improve intervention strategies and measures, conduct interdisciplinary research and learn from the latest advanced model research experience regarding the relationship between the built environment and older adults' PAB.

## LIMITATIONS AND CONTRIBUTIONS

Some limitations of this study should be clarified. Firstly, other elements of the built environment, such as the number of street intersections, accessibility of PA facilities or weather conditions, were also widely concerned in the study of the built environment and behaviors, while they were limited to multiple factors and could not be included, which was a pity of this study. Moreover, utilizing enhanced assessment in the context of building environments (*e.g.*, GIS) and physical activity (*e.g.*, accelerometers rather than questionnaires) is poised to steer forthcoming investigations within this domain with greater precision (*You et al., 2023d*, *2023e*). Secondly, the use of Egger linear regression analysis for testing publication bias is more applicable when the number of studies included in the sample is greater than 10. While among all methods for assessing publication bias in meta-analyses of merged OR values, Egger linear regression analysis seems to have the highest statistical efficacy, this method may still potentially weaken the capacity of our study to identify publication bias to some extent. Last but not least, the large span of the included research areas and the disunity of the measurement methods of the built environment were possible sources of increasing the heterogeneity among the studies, and the participants involved in the study were also limited (*Pham et al., 2020*). In the future, more studies are needed to explore the effects of the built environment on PAB in older adults.

However, it should be underscored that this study was a pioneering effort that has comprehensively searched for studies on the relationship of the built environment and the PABs of older adults. Simultaneously, our meta-analysis provided a broad perspective on current hotspots in this field. It quantitatively reflected the research status in this domain. Our key findings, including the significant influence of factors such as traffic safety, destination accessibility, aesthetics and greenery of sports venues, and building density on older adults' PABs, offer fresh insights into how the built environment can promote PA among older adults. Furthermore, our findings suggested no apparent association between mixed land use, quality of pedestrian or commercial facilities, and older adults PABs, pointing to areas that might require less focus in initiatives aimed at enhancing PA in this age group. By highlighting these insights, we anticipate our study will inspire future research in this field, and contribute to strategies aimed at enhancing the well-being of older adults through PA.

## CONCLUSIONS

Most factors of the built environment might have a positive effect on promoting the development of physical activity behaviors of older adults. These factors which ranked from high to low according to their impact were traffic safety, destination accessibility, aesthetics of sports venues, virescence of sports venues, building density. It seemed that there was no potential association between mixed land use, the quality of pedestrian facilities or commercial facilities and PAB of older adults. The government should improve the built environment and promote physical activity of the aged, and strengthen localization research. Moreover, future studies should strengthen the analysis of the mechanisms of the relationship between the built environment and PA in combination with improving the research quality, collecting multiple research evidences and enhancing interdisciplinary integration.

### Funding
The authors received no funding for this work.

### Competing Interests
The authors declare that they have no competing interests.

### Author Contributions
- Yanwei You conceived and designed the experiments, performed the experiments, analyzed the data, prepared figures and/or tables, authored or reviewed drafts of the article, and approved the final draft.
- Yuquan Chen conceived and designed the experiments, performed the experiments, analyzed the data, prepared figures and/or tables, authored or reviewed drafts of the article, and approved the final draft.
- Qi Zhang conceived and designed the experiments, performed the experiments, analyzed the data, prepared figures and/or tables, and approved the final draft.
- Xiaojie Hu performed the experiments, prepared figures and/or tables, and approved the final draft.
- Xingzhong Li performed the experiments, prepared figures and/or tables, and approved the final draft.
- Ping Yang performed the experiments, prepared figures and/or tables, and approved the final draft.
- Qun Zuo analyzed the data, authored or reviewed drafts of the article, project administration, and approved the final draft.
- Qiang Cao analyzed the data, authored or reviewed drafts of the article, supervision, and approved the final draft.

### Data Availability
  This is a systematic review.
## Supplemental Information

Supplemental information for this article can be found online at http://dx.doi.org/10.7717/peerj.16173#supplemental-information.

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
