# Peer review of "Systematic and meta-based evaluation of the relationship between the built environment and physical activity behaviors among older adults"

_PeerJ, doi:10.7717/peerj.16173_

## Round 0.1 · original submission · Major Revisions

Thank you for your submission. The reviewers have identified a number of concerns that must be addressed. In particular related to the use of the SEM and analysis.

Reviewer 1 ·

Basic reporting

Abstract
1. This is a relationship study, not an effect. Therefore, it is suggested that the authors' title and purpose need to be adjusted.
2. What do you mean by localized research? Is it advice for Chinese people? Advice is not enough if it is only for the Chinese. Because PeerJ is an international journal.

Introduction
1. Line 55. In general, LPA is an abbreviation for light physical activity, not lack of physical activity. It is therefore recommended that the authors revise the LPA throughout the article.
2. The author spends a great deal of time in the second paragraph introducing the social ecology model. Although the model includes elements of the built environment, the built environment is only one of many elements, so using the social ecology model as a theoretical underpinning for the study is clearly not sufficient.
3. Line 100. What do the 5Ds mean? Does it refer to the initials of the 5 indicators?
4. Line 111. The authors point out inconsistencies in the relationship between the built environment and physical activity in older people. Can you give one or two examples of the specific inconsistencies?
5. The specific significance and value of this study is not well distilled in the introduction, and it is suggested that the significance of the study be described in the final paragraph.

Experimental design

Methodology
1. Line 118. There is a misrepresentation. Please replace "Meta-based analysis" with "Meta-analysis".
2. We found the title of your article and the objective of your study to explore the effects of the built environment on physical activity in older people, but why did you retrieve a cross-sectional study? A cross-sectional study is a study that explores the relationship between two variable pieces, not a causal relationship.
3. Line 131. Why did the search start in 2010? Will relevant studies prior to 2010 be missed and will this result in inaccurate findings?
4. Line 133. There are search terms for built environment that appear to be inadequate and it is recommended that similar literature be consulted to add relevant search terms and to retrieve relevant literature.
5. Line 152-153. Please elaborate on why literature with a sample size of less than 100 should be excluded.
6. Line 156-161. The description in this section does not appear to be a data extraction, but rather a literature screening process and results.
7. Line 186-191. Cross-sectional studies can only indicate whether there is a correlation between the built environment and physical activity in older people, not whether there is a facilitative or inhibitory effect.

Validity of the findings

Results
1. The 'Study and Sample Characteristics' can be divided into two parts, the first of which is the 'Literature Search Results' and the second is the 'Data Extraction Results'.
2. The Literature Search Results (Figure 1) section should indicate the search results for each database.
3. Line 269. The results of the combined effects test and the results of the publication bias test are two different things and therefore it is not recommended that they be combined for writing.
4. The authors used Egger linear regression analysis for publication bias testing. However, it is important to note that Egger linear regression analysis has certain sample size requirements. Generally a sample size of 10 or more is required to perform regression analysis.
5. The authors seem to be missing a sensitivity analysis.

Additional comments

Discussions
1. The authors are to be commended for their detailed discussion of the findings of the study. However, it is important to note that the authors cannot use SEM as a starting point for their discussion. This is because the built environment is only a small part of the social ecology.
2. The authors compare the consistency of the findings of the Meta-analysis with those of previous studies for the purpose of explaining their own views. However, the analysis of the mechanisms of the relationship between the built environment and physical activity is missing. For example, how the built environment acts on the human psyche and thus promotes physical activity.
3. What is the usefulness of the main findings of this paper? What are the implications of its research for guiding physical activity? This point needs further clarification by the authors.

Reviewer 2 ·

Basic reporting

The manuscript is clear and unambiguous. The language used is professional. The background information is enclosed sufficiently. The authors claimed that the built environment had a significant effect on the physical activity behaviors of the elderly. Meanwhile, they suggested that more attention should be paid to traffic safety, destination accessibility, aesthetics of sports venues, virescence of sports venues, and building density when promoting physical activity levels among elderly individuals.
However, some small format problems still exist in the manuscript.
The title and content of Table 1 is not displayed properly in the PDF file.
The fonts and layout of table 1 and 2 are not consistent.
It would be better if the authors can put the legend and the title of Figure 2 and 3 larger so the readers can read the figures easily.

Experimental design

The experimental design is enclosed in detail in Figure 1 and the methodology section.

Validity of the findings

The conclusions are well-stated and meaningful for application and further research.
It would be better if the authors can give more detailed examples of the factors related to promoting physical activity levels among elderly individuals.

---

## Round 0.2 · Minor Revisions

It is not appropriate to use the socio-ecological model as the built environment is only a smaller part of the model. Also the model is used for chronic disease or health-related indicator. Please make the necessary corrections.

The manuscript needs to be proofread and edited to make it clearer. In addition, please use "older adults" instead of "elderly". In the Methodology section, use initials for the reviewers instead of their full names. For in-text references, provided only the author's surname.

**Language Note:** The Academic Editor has identified that the English language must be improved. PeerJ can provide language editing services - please contact us at [email protected] for pricing (be sure to provide your manuscript number and title). Alternatively, you should make your own arrangements to improve the language quality and provide details in your response letter. – PeerJ Staff

Reviewer 1 ·

Basic reporting

Dear Author.
I think you have basically met the basic requirements for PeerJ publication with this article. However, I still feel that the socio-ecological model is not applicable to this study. First, the model is a larger concept, while the built environment is only a smaller factor, and second, the socioecological model is applicable to any chronic disease or health-related indicator. Therefore, I still recommend that you make the modifications.

Experimental design

no comment

Validity of the findings

no comment

Reviewer 2 ·

Basic reporting

The authors have replied and made changes to the points in the previous review properly.

Experimental design

The authors have replied and made changes to the points in the previous review properly.

Validity of the findings

The authors have replied and made changes to the points in the previous review properly.

---

## Round 0.3 · accepted · Accept

Thank you for your revised submission. I am satisfied that you have addressed the remaining concerns, and am happy to accept your paper for publication.